# Quality of informal care among informal caregivers of people with dementia: A latent profile and ROC analysis

Chan Cai[1,2‡], Bing Cheng[1‡], Chongqing Shi [1*], Wenli Shi[1], Chenyang Li[1], Cui Liu[1], Jin Sun[3]

1 Institute of Nursing Research, Hubei Province Key Laboratory of Occupational, Hazard Identification and Control, School of Medicine, Wuhan University of Science and Technology, Wuhan, Hubei, China, 2 Department of Nursing, Xiangyang No.1 People's Hospital, University of Medicine, Xiangyang, Hubei, China, 3 Yiling Hospital of Yichang City, Yichang, Hubei, China

‡ CC, BC have contributed equally to this work.
* shichongqing@wust.edu.cn

## Abstract

### Background

The quality of informal care for people with dementia (PwD) has gained increasing importance, as most PwD prefer home-based care over institutional placement. However, evidence-based intervention programs tailored to distinct care quality profiles remain limited. Additionally, the absence of clear thresholds to identify PwD receiving low-quality informal care poses a challenge for research and clinical practice. Thus, this study aimed to identify the profiles of quality of care (QoC) among informal caregivers of PwD, explore influencing factors of different profile, and determine the optimal cut-off score of the Exemplary Care Scale (ECS).

### Methods

A cross-sectional survey was conducted. A total of 213 dyads of PwD and their informal caregivers were recruited from memory clinic, rehabilitation clinic, and neurological clinic of a tertiary hospitals and communities in Wuhan, Hubei, China, between July 15, 2023, and July 14, 2024. Latent profile analysis (LPA) was employed to identify QoC profiles. Multinomial logistic regression was performed to explore influencing factors of profile membership. Receiver Operating Characteristic (ROC) analysis was conducted to determine the ECS cut-off score.

### Results

Three distinct QoC profiles were identified: high (24.41%), moderate (44.60%), and low (30.99%). Among informal caregivers, lower monthly income, insufficient social support, and higher perceived overload were associated with low QoC profile, whereas, better quality of pre-illness relationship with PwD and greater activities of

**Data availability statement:** All relevant data are within the manuscript and its Supporting Information files.

**Funding:** The author(s) received no specific funding for this work.

**Competing interests:** The authors have declared that no competing interests exist.

**Abbreviations:** QoC: quality of care; PwD: people with dementia; ECS: Exemplary Care Scale; RRS: Relationship Rewards Scale; BI: The Barthel Index; CES-D: Epidemiologic Studies Depression Scale; SSRS: Social Support Rating Scale; LPA: Latent profile analysis; ADL: activity of daily living; CRs: care recipients; CGs: caregivers; ref: reference; CI: confidence interval.

daily living (ADL) of PwD were associated with high QoC. ROC analysis yielded an optimal ECS cut-off score of 15, with high sensitivity (0.993) and specificity (0.955).

## Conclusions

This study identified three distinct QoC profiles among caregivers of PwD, underscoring the heterogeneity of informal care quality. The identified predictors and the validated ECS cut-off score of 15 provide an empirical basis for developing tailored screening tools and targeted interventions for high-risk caregiver subgroups.

## 1. Introduction

Dementia is a syndrome characterized by progressive cognitive impairment that is associated with marked declines in patient's ability to perform activities of daily living (ADL), learning, work, and social interactions [1]. Globally, the number of people with dementia (PwD) is projected to reach 152 million by 2050 [2]. Currently, one in four PwD reside in China, which ranks first worldwide in terms of number of cases [3]. Most PwD live at home and are cared for by informal caregivers (CGs), typically spouses or children [4]. This care is based both on the cultural tradition of caring for elders and on a greater knowledge of their preferences and values than formal caregivers may have or provide. However, sustained caregiving often generates considerable stress and burden, along with a range of physical and mental health issues for caregivers [5], which can in turn undermine the quality of care (QoC) provided [6,7]. In this sense, some studies have shown that low QoC adversely affects the health of PwD, increasing the risk of recurrent hospitalizations and institutionalization, outcomes that can ultimately affect their life expectancy [8,9]. Thus, early identification of caregivers at risk of providing low-quality care and the implementation of targeted support are urgently needed.

QoC is a multidimensional construct encompassing the assessment of key caregiving aspects. These include behaviors that may harm care recipients' (CRs) well-being, the adequacy of caregiving behaviors in meeting the CRs' basic needs, and responsiveness to CRs' psychological needs for respect and enjoyable social engagement [10]. Traditionally, QoC assessment has focused on identifying harmful caregiver behaviors and assessing the extent to which CRs' basic and instrumental daily needs are met [10]. However, these dimensions are not sensitive to the feelings and wishes of CRs and, therefore, fail to reflect all the aspects that quality care for PwD living in the community should consider. To address this gap, Dooley et al. conducted an extensive literature review and proposed the concept of exemplary care [11]. Exemplary care extends beyond meeting basic needs to include respecting CRs' feelings, preferences, opinions, and values, while actively avoiding actions that criticize or undermine their dignity. It is fundamentally characterized by sensitivity and respect demonstrated throughout the caregiving process. This approach conveys to CRs the message that they are loved, respected, and worthy of special attention, helping to maintain their dignity and integrity, and holistically reflecting the concept of person-centered care.

In this line, and based on the conceptual framework of exemplary care, Dooley and colleagues developed the Exemplary Care Scale (ECS) to evaluate the performance of informal caregivers in home-based dementia care [11]. The scale comprises two core domains: care provision and respect demonstration. Compared to other instruments commonly used to assess informal care quality, such as the comprehensive but lengthy QUALCARE Scale [12], the functionally-oriented Family Caregiving Consequences Inventory (FCCI) [13], and the negatively-focused Potential Harmful Behaviors scale (PHB) [14,15], the ECS stands out with its unique positive perspective on caregiving. Moreover, its cross-cultural validity has been confirmed through rigorous validation studies in diverse populations, including the United States [16], Argentina [17], and China [18]. These attributes make the ECS a particularly reliable and practical tool for assessing QoC for PwD in Chinese household settings.

Although much research focuses on the total ECS score to assess overall quality of care and the factors that influence it [16–20], this variable-centered approach fails to capture individual differences and does not allow for the observation of heterogeneity across subgroups. A shift to a person-centered approach is therefore needed to identify distinct caregiver profiles for personalized support. Although such person-centered methods have identified meaningful profiles in elderly care [21], their application specifically to dementia caregivers remains scarce. This gap limits the empirical basis for providing targeted support to high-risk caregivers who provide low QoC for PwD.

On the other hand, although heterogeneity analysis effectively captures differences among groups [22], its utility for rapid clinical screening is limited. To address this practical challenge, we propose to investigate in greater depth through this study the optimal cut-off score of ECS using Receiver Operating Characteristic (ROC) analysis. This approach would allow clinicians and community professionals to quickly identify high-risk groups and establish action strategies, using only the total score as a reference. The proposed simplification of the scoring process represents a methodological innovation that preserves the advantage of the LPA in identifying heterogeneity and improves its clinical applicability, while also providing empirical evidence for understanding the causes of this heterogeneity.

In this context, our hypotheses are: (1) the quality of informal care could be categorized into different latent profiles according to the ECS; (2) the quality of informal care provided by caregivers varies depending on individual characteristics, stressors, quality of life, and social support. (3) On this basis, and through ROC analysis, it may be possible to establish an optimal cutoff score with high sensitivity and specificity that allows for the identification of different levels of care quality and, specifically, the identification of low-quality care.

These results would have two practical implications. First, they would provide a scientific basis for developing targeted interventions tailored to different subgroups. Second, they could significantly improve the assessment of care quality and, thereby, efficiency.

To respond to these hypotheses, we set four objectives: (1) to identify potential subgroups of the quality of care among informal caregivers of PwD,; (2) to clarify the factors associated with these subgroups; (3) to identify categorization thresholds for the ECS that can be applied in clinical and community practice; and (4) to propose rapid screening criteria that enable health and social organizations to quickly identify high-risk individuals.

This study was guided by McClendon's Extended Stress Process Model [16]. The model takes "QoC" as the core outcome variable, shifting the research perspective from "caregiver-centered" to the "caregiver-patient dyadic integration." Including multidimensional influences provides a systematic theoretical framework for understanding QoC in dementia. The results of this study could provide important empirical evidence for the subsequent development of precise intervention strategies for different subgroups of caregivers.

## 2. Materials and methods

### 2.1 Design and participants

This cross-sectional study was conducted between July 15, 2023, and July 14, 2024, in Wuhan, Hubei Province, China. Using convenience sampling, we recruited PwD and their informal caregivers attending memory clinic, rehabilitation clinic, and neurological clinic in the hospital and communities. PwD were eligible if they: (1) were aged ≥60 years;(2) met the

                                                                                

WHO International Classification of Diseases, 10th Revision (ICD-10) diagnostic criteria for dementia [23]; and (3) had lived at home with family for at least three months prior to enrollment.

All caregivers were primary informal caregivers of PwD who were unpaid and met the following conditions: (1) aged ≥18 years, (2) providing at least four hours per day on caregiving for no less than three months, thereby fulfilling the role of the primary caregiver [24]; (3) often accompanying patients to see a doctor, the best understanding of the patient's condition, and basic living conditions; (4) provide informed consent; and (5) were able to read, comprehend the Chinese questionnaire, and communicate effectively in Mandarin (the official language of China). Caregivers were excluded if they had experienced other major stressful life events (e.g., bereavement or divorce) within the past three months. All participants were assessed by researchers to ensure they understood the study.

## 2.2 Data collection

Data were collected using paper-based questionnaires administered during face-to-face interviews with caregivers. Caregivers served as the sole respondents, providing information on both their own status and that of the PwD. After providing written informed consent, participants completed the questionnaires anonymously. Participants with reading difficulties completed the questionnaires with the help of trained research staff. The questionnaires were completed and collected on-site. After collecting questionnaires, the investigators immediately checked the questionnaires for missing or obvious logical errors, and any issues were resolved on the spot. Each questionnaire was completed within 15–25 minutes.

## 2.3 Sample size calculation

An a priori sample size calculation was performed using G*Power 3.1.9.7. With a two-tailed, a medium effect size ($f^2 = 0.30$), a power of 0.95, a statistical level $\alpha = 0.05$ [25], and 20 predictors, the minimum required sample size was 120. To ensure robust model fit, stable parameter estimation, and adequate power for subgroup comparisons in LPA, we targeted a minimum of 200 participants [26], consistent with methodological recommendations for person-centered approaches. Initially, 248 caregiver-PwD dyads were recruited. After excluding 35 dyads due to incomplete responses or other reasons, 213 valid dyads were included in the final analysis.

## 2.4 Ethics approval

The study was approved by the Ethics Committee of TianYou Hospital Affiliated to Wuhan University of Science &Technology (LL-2023-07-14-002).

## 2.5 Measures

All information was obtained from informal caregivers, covering two primary domains. The first domain comprised sociodemographic and clinical details of PwD, including age, gender, education level (1= ≤ 6 years; 2 = 7–9 years; 3= ≥ 10 years), type of dementia, years since diagnosis, number of children, ADL, and residence. The second domain encompassed caregivers' own characteristics: age, gender, education level, relationship with PwD (1 = children; 2 = partner; 3 = other), cohabitation status (1 = yes; 2 = no), length of caregiving, monthly income (RMB) (1= ≤ 2000; 2 = 2000–5000; 3= ≥ 5000), and physical health. Physical health was assessed with the single-item question, "How do you rate your general health?", rated on a 5-point scale from very poor to very good. For analysis, responses were dichotomized as impaired (1 = very poor, poor, fair) and good (2 = good, very good). Additionally, caregivers completed five instruments assessing quality of care, perceived overload, quality of the pre-illness relationship with PwD, depressive symptoms, and social support.

**2.5.1 Activities of daily living (ADL).** The Barthel Index (BI), developed by Florence Mahoney and Dorothy Barthel [27], is a 10-item scale used to assess patients' ADL. Total scores range from 0 to 100, with higher scores indicating

greater independence. Based on the BI score, patients' ADL is classified into four levels: independence (100), mild dependence (65–95), moderate dependence (40–60), and severe dependence (0–35) [28]. Data were collected via face-to-face interviews conducted by trained researchers with family caregivers.

**2.5.2 Quality of care (QoC).** QoC was measured using the Exemplary Care Scale (ECS), which was translated into Chinese by Lau et al [18]. The ECS was completed by the informal caregivers. It consists of 11 items that fall into two dimensions: Provide (items 1–5) and Respect (items 6–11). Each item is rated on a 4-point scale (0 = never, 1 = sometimes, 2 = often, and 3 = always). The total score ranges from 0 to 33. Higher scores indicate better QoC. Cronbach's alpha for the entire scale was 0.816.

**2.5.3 Quality of pre-illness relationship.** The Relationship Rewards Scale (RRS) was used to assess the quality of the pre-illness relationship between caregiver and CR [15]. Caregivers assessed the frequency (1 = never, 2 = sometimes, 3 = often, 4 = always) at which they perceived pre-illness interpersonal relationships with CRs as rewarding, with higher scores indicating better quality of pre-illness relationship. The RRS was translated into Chinese using Brislin's guidelines [29]. Cronbach's alpha in this sample was 0.840.

**2.5.4 Perceived overload.** Caregiver perceived overload was assessed using the 4-item Overload Scale. Items were rated on a 4-point scale (1 = not at all, 2 = somewhat, 3 = quite a bit, 4 = completely), with total scores ranging from 4 to 16, with higher scores indicating a higher perceived overload [30]. The Chinese version was developed following Brislin's guidelines [29]. Cronbach's alpha of the Overload scale was 0.791.

**2.5.5 Depression.** Caregivers' depression was assessed using the Center for Epidemiologic Studies Depression Scale (CES-D) developed by Radloff [31]. The 20-item CES-D scale asks respondents how often they had experienced depressive symptoms in the past week. A cut-off score of ≥16 was used to indicate clinically significant depression [32]. Cronbach's alpha for the CES-D was 0.877 in this study.

**2.5.6 Social support.** Social support was evaluated using the 10-item Social Support Rating Scale (SSRS), which was designed for the Chinese population by Xiao, based on the Social Support Questionnaire (SSQ) and Interview Schedule for Social Interaction (ISSI) [33,34]. The SSRS comprises three dimensions: subjective support, objective support, and support utilization. Total scores range from 12 to 66 [35], and are categorized into three levels: low-level (≤22), medium-level (23–44), and high-level (≥45). In this study, Cronbach's alpha for the SSRS was 0.804.

## 2.6 Statistics analysis

We conducted four statistical analyses. First, descriptive statistics were used to summarize participant characteristics. Second, LPA was performed using Mplus 8.3 to identify distinct QoC profiles, with each item of the ECS serving as an indicator. Model fit was evaluated using multiple indices: Akaike Information Criterion (AIC), Bayesian Information Criterion (BIC), and adjusted Bayesian Information Criterion (aBIC), Lo-Mendell-Rubin likelihood ratio test (LMR), Bootstrapped likelihood ratio test (BLRT), and Entropy [36]. Lower AIC, BIC, and aBIC values indicated better fit. Differences between latent profile models were compared using the LMR and BLRT. Significant LMR and BLRT *P*-value suggested that a *k* class model fit significantly better than a *k*-1 class model. Entropy ≥0.8 indicates a classification accuracy exceeding 90%. Cohen's d was calculated to further validate the accuracy of the classification (0.2–0.5: small; 0.5–0.8: medium; >0.8: large) [37,38]. Third, to examine the effects of predictor variables on latent profiles of QoC, we employed the R3STEP command in Mplus, which effectively controls for potential effects arising from classification errors [39]. Fourth, ROC analysis was performed to determine the optimal cut-off value for the ECS using SAS 9.4. The optimal threshold was defined as the value that maximized Youden's index (sensitivity + specificity–1), with higher values indicating better discriminatory performance [40]. The area under the curve (AUC) was computed to evaluate the accuracy of identifying low QoC. AUC values were interpreted as follows: ≥ 0.90 (excellent), 0.80–0.89 (good), 0.70–0.79 (fair), and <0.70 (poor). The *P*-value <0.05 indicated statistical significance.

# 3. Results

## 3.1 Sample characteristics

A total of 213 dyads of PwD and their informal caregivers were recruited. The mean age of PwD was 77.29±9.42 years, and most were cared for by spouses or children. Notably, 54.9% of the PwD were male. Caregivers were predominantly female (64.8%) and 75.6% lived with PwD. Further details are presented in Table 1. Additionally, Table 2 summarizes the scores of key caregiver variables, including quality of pre-illness relationship, perceived overload, and social support, which were used in subsequent analyses.

**Table 1. General characteristics of PwD and caregivers (N=213).**

| PwD | M±SD or N (Percentage) | Informal caregivers | M±SD or N (Percentage) |
|---|---|---|---|
| Age (years) | 77.29±9.42 | Age(years) | 59.06±14.33 |
| Gender | | Gender | |
| Male | 117 (54.9) | Male | 75 (35.2) |
| Female | 96 (45.1) | Female | 138 (64.8) |
| Education level | | Education level | |
| ≤6 years | 112 (52.6) | ≤6 years | 45 (21.1) |
| 7-9 years | 87 (40.8) | 7-9 years | 94 (44.1) |
| ≥10 years | 14 (6.6) | ≥10 years | 74 (34.8) |
| Type of dementia | | Relationship with PwD | |
| Alzheimer | 128 (60.1) | Spouse | 76 (35.7) |
| Vascular dementia | 71 (33.3) | Children | 110 (51.6) |
| Other | 14 (6.6) | Other[a] | 27 (12.7) |
| Years since diagnosis | | Cohabitation status | |
| ~1 | 58 (27.2) | Yes | 161 (75.6) |
| ~5 | 99 (46.5) | No | 52 (24.4) |
| ≥5 | 56 (26.3) | Length of care (year) | |
| Numbers of children | | ~1 | 70 (32.9) |
| 0 | 1 (0.5) | ~5 | 97 (45.5) |
| 1 | 50 (23.5) | ≥5 | 46 (21.6) |
| 2 | 70 (32.8) | Monthly income (RMB) | |
| ≥3 | 92 (43.2) | ≤2000 | 62 (29.1) |
| ADL | | 2000~5000 | 67 (31.5) |
| Severe | 101 (47.4) | ≥5000 | 84 (39.4) |
| Moderate | 36 (16.9) | Physical health | |
| Mild | 59 (27.7) | Poor | 140 (65.7) |
| No | 17 (8.0) | Good | 73 (34.3) |
| Residence | | Depression | |
| Rural | 70 (32.9) | No | 144 (67.6) |
| Urban and town | 143 (67.1) | Yes | 69 (32.4) |

Note: PwD, people with dementia; ADL, activities of daily living; Other[a], included siblings, friends, and other relatives; M, mean; SD, standard deviation; RMB, Ren Min Bi (the Chinese yuan, ¥).

**Table 2. Scores for caregivers' quality of pre-illness relationship, perceived overload, and social support (N=213).**

| Variables | M±SD |
|---|---|
| Quality of pre-illness relationship | 11.68±2.82 |
| Perceived overload | 9.63±2.90 |
| Social support | 39.25±7.04 |

Note: M, mean; SD, standard deviation.

## 3.2 Latent profile analysis

Fit indices for the 1- to 5-profile solutions are shown in Table 3. The AIC, BIC, and aBIC values decreased continuously as the number of profiles increased. The 4-profile model was excluded because the LMR *P*-value was not significant (*P*>0.05). For the 2-, 3-, and 5-profile models, both LMR and BLRT *P*-values were significant, indicating that each *k* profiles provided a significantly better fit than *k*-1 profiles model. Although the 5-profile model yielded lower AIC, BIC, and aBIC values and higher entropy than the 3-profile model, the additional profile contributed limited substantive information and resulted in smaller, less interpretable subgroups. Considering both statistical fit and parsimony, the 3-profile model was selected as the optimal model. The average probability of QoC belonging to the profile ranged from 92.0% to 95.2% (Table 4), indicating excellent classification accuracy. All Cohen's d values for pairwise comparisons between profiles exceeded 0.80 (Table 5), further supporting the distinctiveness of the identified subgroups.

The three latent profiles exhibited distinct scoring patterns across the eleven items of ECS. Profile 1 had the highest mean scores on all items and was therefore labeled the high QoC profile, comprising 24.4% of the sample. Profile 2 showed moderate scores on all items and was designated the moderate QoC profile, accounting for 44.6%. Profile 3 had the lowest scores on each item and was termed the low QoC profile, representing 31.0% of the caregivers. The scores of the three profiles on the eleven item of QoC are depicted in Fig 1.

## 3.3 Influencing factors analysis

After univariate analyses identified significant variables (S1 Table), multinomial logistic regression using the R3STEP procedure in Mplus was conducted to examine their effects on profile membership, with the moderate QoC profile as the reference. Results are presented in Table 6.

Lower monthly income (*β*=−1.359, *P*<0.05), insufficient social support (*β*=−0.206, *P*<0.01), and higher perceived overload (*β*=0.359, *P*<0.01) were significantly associated with low QoC profile. In contrast, better quality of pre-illness relationship (*β*=0.316, *P*<0.01) and and better ADL function of the PwD (*β*=1.070, *P*<0.001) were associated with high QoC profile.

**Table 3. Fit information of the latent profile models.**

| profile | Likelihood | AIC | BIC | aBIC | LMR(*P*) | BLRT(*P*) | Entropy | Proportion |
|---|---|---|---|---|---|---|---|---|
| 1 | −2596.885 | 5237.769 | 5311.718 | 5242.006 | – | – | – | – |
| 2 | −2370.772 | 4809.544 | 4923.828 | 4816.092 | 0.000 | 0.000 | 0.892 | 0.338/0.662 |
| **3** | **−2289.612** | **4671.224** | **4825.844** | **4680.083** | **0.016** | **0.000** | **0.855** | **0.244/0.446/0.310** |
| 4 | −2250.044 | 4616.088 | 4811.043 | 4627.258 | 0.517 | 0.000 | 0.851 | 0.300/0.174/0.235/0.291 |
| 5 | −2017.855 | 4283.771 | 4519.061 | 4297.251 | 0.017 | 0.000 | 0.956 | 0.146/0.150/0.113/0.221/0.371 |

Note: AIC, Akaike Information Criterion; BIC, Bayesian Information Criterion; aBIC, adjusted Bayesian Information Criterion; LMR, Lo-Mendell-Rubin likelihood ratio test; BLRT, Bootstrapped likelihood ratio test.

**Table 4. Probability of different categories in the latent profile analysis.**

| Class | Profile 1 | Profile 2 | Profile 3 |
|---|---|---|---|
| Profile 1 | 0.952 | 0.000 | 0.048 |
| Profile 2 | 0.000 | 0.920 | 0.080 |
| Profile 3 | 0.028 | 0.046 | 0.927 |

**Table 5. Means, standard deviations, and Cohen's d for the three profiles of Exemplary Care Scale.**

| | M(SD) | N(%) | Score Ranges | Cohen's d |
|---|---|---|---|---|
| Low QoC | 11.35 (2.10) | 66 (30.99%) | [5, 16] | $d_{2-1} = 3.24$ |
| Moderate QoC | 18.13 (2.09) | 95 (44.60%) | [12, 22] | $d_{3-2} = 2.06$ |
| High QoC | 22.73 (2.51) | 52 (24.41%) | [19, 29] | $d_{3-1} = 4.97$ |

Note: QoC, quality of care; M, mean; SD, standard deviation; N(%): Number (Percentage). Cohen's $d_{2-1}$: the standardized mean difference between moderate QoC group and low QoC group; Cohen's $d_{3-1}$: the standardized mean difference between high QoC group and low QoC group; Cohen's $d_{3-2}$: the standardized mean difference between high QoC group and moderate QoC group.

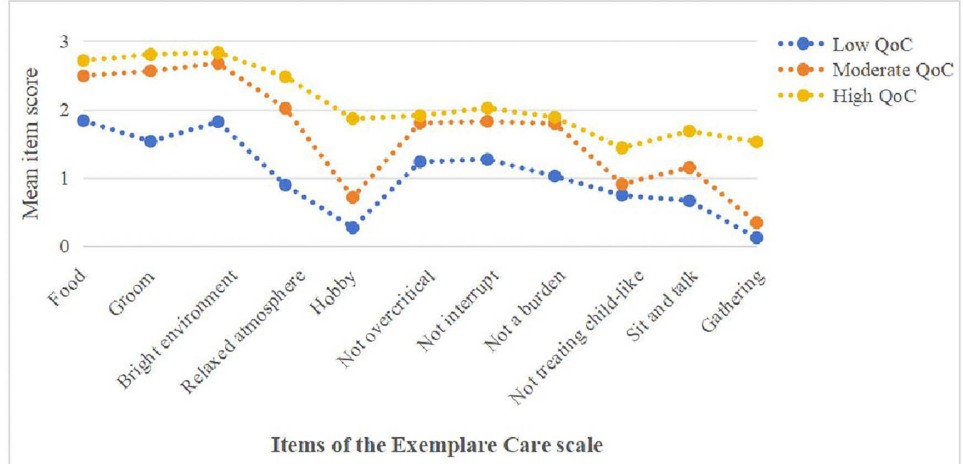

**Fig 1. Mean item scores across the three-profile model of the Exemplary Care Scale. The figure displays the mean scores of each item across the three profiles.**

### 3.4 ROC analysis

To validate the clinical applicability of the ECS cut-off, the sample was randomly split into two independent subsamples (approximately 50%/50%). In the derivation subsample, ROC analysis identified an optimal ECS cut-off score of 15, yielding a high sensitivity of 0.993, specificity of 0.955, and an AUC of 0.998 (95% CI [0.993, 0.999], $P < 0.001$; Fig 2; Table 7). The mean difference between the two groups was statistically significant with a large effect size (Table 8), further supporting this cut-off point. In the validation subsample, the identical cut-off of 15 was confirmed, achieving a Youden's index of 0.928, sensitivity of 0.985, specificity of 0.943, and an AUC of 0.990 (95% CI [0.975, 0.999], $P < 0.001$; S2 Table; S1 Fig). The consistent performance across both subsamples supports the robustness and generalizability of the ECS cut-off score of 15 for identifying low QoC.

**Table 6. Results of multinomial logistic regressions predicting profile membership (R3STEP) (N = 213).**

| Variable | low VS moderate (ref) | | high VS moderate (ref) | |
|---|---|---|---|---|
| | β (SE) | OR (95% CI) | β (SE) | OR (95% CI) |
| **background variables** | | | | |
| Monthly income (RMB) | −1.359* (0.667) | 0.257(−2.666, −0.053) | −0.221 (0.481) | 0.802 (−1.164, 0.722) |
| Quality of pre-illnes relationship with PwD | −0.235 (0.133) | 0.790(−0.496, 0.025) | 0.316** (0.119) | 1.372 (0.082, 0.550) |
| **stressors** | | | | |
| PwD's ADL | 0.090 (0.358) | 1.094(−0.612, 0.791) | 1.070** (0.309) | 2.914 (0.464, 1.675) |
| Caregivers' perceived overload | 0.359** (0.134) | 1.431(0.095, 0.622) | 0.066 (0.136) | 1.068 (−0.202, 0.333) |
| **caregiver resource** | | | | |
| Social support | −0.206** (0.065) | 0.814(−0.333, −0.079) | 0.070 (0.056) | 1.072 (−0.040, 0.179) |

Note: PwD, people with dementia; ADL, activity of daily living; OR, Odds Ratio; 95%CI, 95% Confidence Interval; ref, reference; RMB, Ren Min Bi (the Chinese yuan, ¥); SE, Standard Error; *P < 0.05; **P < 0.01; ***P < 0.001.

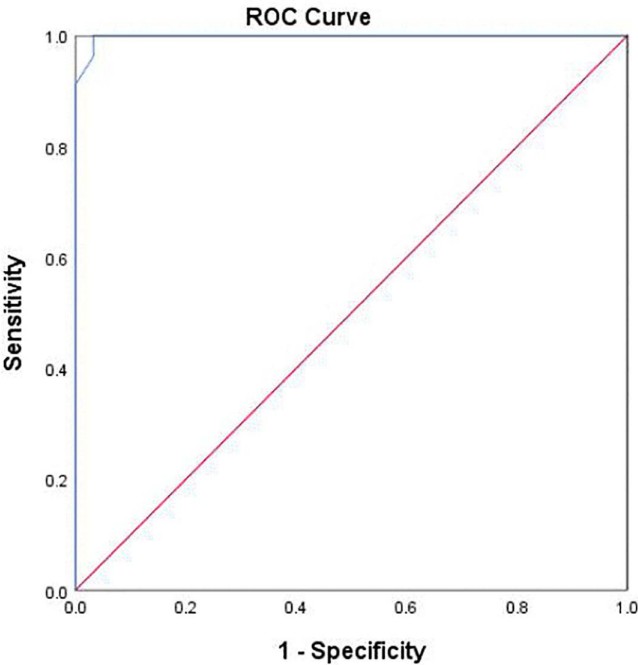

**Fig 2. ROC curve of the Exemplary Care Scale for classifying low and high quality of care.**

## Discussion

This study identified three distinct profiles of informal care quality among caregivers of PwD, confirming significant heterogeneity in the quality of informal care. The low QoC profile scored the lowest on items related to emotional aspects (e.g., Hobby, Gathering), reflecting a pattern of survival-oriented, passive care that addressed only basic life needs. Social engagement, emotional support, and respectful treatment were largely absent, indicating minimal attention to the PwD's dignity, preferences, or psychosocial well-being [18]. The low QoC profile was driven by a convergence of caregiver

**Table 7. Criterion values and coordinates of ROC Curve (derivation subsample).**

| Cut-off point | Sensitivity | Specificity | Youden's index |
|---|---|---|---|
| 14 | 1.000 | 0.839 | 0.839 |
| *15* | **1.000** | **0.968** | **0.968** |
| 16 | 0.962 | 0.968 | 0.930 |
| 17 | 0.911 | 1.000 | 0.911 |

Note: Estimates in italics are the suggested optimal cut-off points; ROC: Receiver Operating Characteristic.

**Table 8. Negative and positive groups classified by the optimal cut-off score of the Exemplary Care Scale.**

| | M(SD) | N(%) | Score Ranges | Cohen's d |
|---|---|---|---|---|
| Negative group | 11.10 (1.96) | 31 (28.18%) | [5,15] | d = 3.49 |
| Positive group | 19.85 (2.69) | 79 (71.82%) | [12,29] | |

Note: M, Mean; SD, Standard Deviation; N(%): Number (Percentage). Cohen's d: the standardized mean difference between the Negative group and the Positive group;

adversities: economic strain, high overload, and inadequate support, highlighting structural determinants of poor outcomes. This profile may be particularly common among caregivers of severely dependent PwD, those with prolonged care duration, or those struggling with limited financial resources to sustain care. The moderate QoC profile, which represented the largest caregivers group, demonstrated adequate performance in daily life care tasks but exhibited insufficient emotional support. While basic life needs were reasonably well met, caregivers showed limited responsiveness to the PwD's psychological needs, individual interests, or social participation. This task-centered approach left considerable room for improvement in overall care quality. In contrast, the high QoC profile exemplified dignity-oriented care. Caregivers not only delivered high quality life support but also actively consider the PwD's emotional needs, autonomy, and social engagement. This profile fully embodied the core principles of comprehensive, high-quality care provision.

These distinct profiles were underpinned by different configurations of background variables, stressors, and resources. Overall, our findings are largely consistent with the Extended Stress Process Model [16], confirming that background variables, primary stressors, and caregiver resources significantly predict the quality of informal care, but did not find a similar impact from caregivers' quality of life. Specifically, contrary to both the model and prior research [7,16,41], caregiver depression, a core dimension of quality of life, did not show a significant association with QoC. This nonsignificant association can be attributed to Chinese cultural norms, particularly filial piety and family responsibility. This cultural value has been linked to reduced caregiver burden and enhanced QoC [42], as home-based dementia care in China is rooted in familism and filial norms [43]. Within this context, family caregivers may find fulfillment in their caregiving role [44], reinforced by community recognition, which enables them to sustain caregiving standards despite experiencing depressive symptoms. This mechanism explains our core finding: although depressive symptoms differed significantly across the three latent profile groups (S1 Table), they did not predict QoC. In other words, depression effectively captured caregiver heterogeneity, yet its impact on caregiving behavior was buffered by cultural responsibility. Thus, the observed nonsignificant association reflects the unique cultural context of family caregiving in China, rather than any inadequacy in scale validity. Future research should develop culturally sensitive models for assessing QoC.

Monthly income was a key factor for QoC [45,46]. Higher-income caregivers can provide more material support during treatment and rehabilitation, contributing to better QoC and patient outcomes. Low education and rural residence were strong predictors of low income [47], thereby indirectly increasing the risk of poor QoC. These findings identify caregivers

with low income, low education, and rural residence as a multiple disadvantaged subgroup, underscoring the need for integrated economic and educational support from social services.

In alignment with previous studies [15,48], a positive pre-illness relationship serves as a protective factor for QoC. As Steadman et al. demonstrated [49], caregivers reporting higher premorbid relationship satisfaction showed significantly less stress and less reactivity to memory and behavioral problems, as well as better problem-solving skills and more effective communication. Conversely, caregivers with poor pre-illness relationships may perceive caregiving responsibilities as an additional burden, leading to negative attitudes and behaviors [46], ultimately compromising the quality of informal care. This finding emphasizes that pre-illness relationship quality, while not modifiable, is a critical factor. It suggests prioritizing interventions for caregivers with poorer pre-existing relationships, as they may require more intensive support to achieve exemplary care.

At the same time, when PwD have a greater ability to perform basic activities of daily living, this situation correlated positively with the quality of care, while caregiver burden showed a negative correlation, results that have also been observed in a large number of studies [7,45,50]. Caregivers of PwD experience high levels of stress and overload in long-term care [51], compounded by competing family and occupational responsibilities. Full-time, long-term care impairs caregivers' mental health and care capacity, further diminishing quality of care [41,52].

In addition to the caregiver-related factors, patient-level characteristic also play a critical role in shaping QoC. Research has shown that cognitive impairment or psychiatric symptoms are prevalent in most patients at the time of diagnosis [3], substantially compromising their daily functioning and necessitating caregiver assistance. However, many caregivers of PwD struggle to provide consistent and attentive care due to insufficient care capacity and a lack of care resources [53,54], ultimately resulting in diminished QoC. This challenge is compounded by the fact that progressive disability is often accompanied by aphasia and comprehension deficits [3], making it difficult for patients to express their needs accurately. These challenges can undermine caregivers' patience and lead to inadequate care [55]. Consequently, as PwD's ADL decline, QoC may also decrease. Therefore, assessing the ability of PwD to perform ADL is an indispensable tool for providing individualized care and for proposing measures to increase these abilities, all with the goal of improving the quality of care for PwD.

The buffering model of social support posits that social support buffers against the adverse impact of stressful events [56]. However, China's current caregiving support system is underdeveloped [57]; caregivers predominantly rely on informal family networks, while formal support at both community and national levels remains insufficient [58]. The current uni-dimensional and incomplete support structure underscores the urgent need to strengthen community- and national-level formal support, thereby building a diversified social support network that can provide caregivers with adequate resources and assistance.

Combining LPA and ROC analysis, the study revealed that a cut-off point of 15 had the highest sensitivity and specificity for identifying individuals with low quality of informal care, offering an evidence-based threshold for future research and practice. This cut-off score enables healthcare professionals to rapidly identify caregivers at high risk of providing inadequate care, prompting timely evaluation and follow-up. For social services planning, this benchmark supports the efficient allocation of resources by prioritizing interventions for the most vulnerable caregiver-PwD dyads. Public health professionals are suggested to collect information on the quality of informal care for PwD, implement corresponding interventions for family caregivers of PwD whose scores are less than 15, and provide necessary support.

## 5. Limitations

First, due to the cross-sectional design, causal relationships cannot be established. Future longitudinal studies are needed to strengthen the conclusions. Second, the sample was restricted to informal caregivers of PwD recruited from a single city in China, which may limit the generalizability of the findings. Future research should expand the sample to caregivers from diverse regions and urbanization levels to enhance external validity. Third, although the ECS cut-off score

of 15 demonstrated high sensitivity and specificity in the ROC analysis, this threshold was derived from a single sample and requires cross-validation in different samples to support its clinical applicability. Fourth, the use of self-reported ECS may be subject to social desirability bias and recall errors. Future studies could incorporate objective measures (e.g., clinical assessments, or behavioral observations) to improve data reliability. Fifth, our sample included both co-residing and non-co-residing caregivers, which may entail distinct caregiving contexts and challenges. Stratifying by residential status in future research would enable a more nuanced understanding of the unique experiences and support needs of each subgroup.

## 6. Conclusion

This study confirmed all three hypotheses. Using LPA, we identified three distinct QoC profiles: low, moderate, and high, demonstrating significant heterogeneity among caregivers of PwD. Guided by McClendon's Extended Stress Process Model, we further found that lower monthly income, inadequate social support, and higher perceived overload significantly predicted low QoC; better pre-illness relationship quality and greater ability to perform ADL of PwD were key predictors of high QoC. These findings underscore the need for public health professionals to comprehensively assess these multidimensional factors. Targeted interventions should be applied simultaneously to heterogeneous informal caregivers to improve care quality and prevent its deterioration. To facilitate early identification, we established an optimal ECS cut-off score of 15 via ROC analysis, which demonstrated excellent sensitivity and specificity. This evidence-based threshold provides a practical screening tool for timely detection and targeted intervention in both clinical and community settings.

## Supporting information

**S1 Table. Univariate analysis of different latent profile (N = 213).**
(PDF)

**S2 Table. Criterion values and coordinates of ROC Curve (validation subsample).**
(PDF)

**S1 Fig. ROC of the Exemplary Care Scale score for classifying low and high QoC.**
(PDF)

**S1 File. Data.**
(XLSX)

## Acknowledgments

We are grateful to all the people who agreed to participate in this study for their support, time, and patience. We also thank all staff in the hospital and community centers for their patience and cooperation.

## Author contributions

**Conceptualization:** Chan Cai, Chongqing Shi.

**Data curation:** Bing Cheng, Chongqing Shi.

**Investigation:** Chan Cai, Bing Cheng, Wenli Shi, Chenyang Li, Cui Liu, Jin Sun.

**Methodology:** Chan Cai, Bing Cheng, Wenli Shi, Chenyang Li.

**Supervision:** Chongqing Shi.

**Writing – original draft:** Chan Cai.

**Writing – review & editing:** Chan Cai, Bing Cheng, Chongqing Shi, Wenli Shi, Chenyang Li, Cui Liu.

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
