## [Decision Letter · Decision Letter 0]

26 Aug 2025

Dear Dr. Shi,

Thank you for submitting your manuscript to PLOS ONE. After careful consideration, we feel that it has merit but does not fully meet PLOS ONE’s publication criteria as it currently stands. Therefore, we invite you to submit a revised version of the manuscript that addresses the points raised during the review process.

We look forward to receiving your revised manuscript.

Kind regards,

Arupendra Mozumdar

Academic Editor

PLOS ONE

2. In the online submission form, you indicated that [The data used or analyzed in this study may be obtained from the corresponding author upon reasonable request.].

Additional Editor Comments:

Based on my review and the suggestions from two reviewers, we invite the authors to revise the manuscript and resubmit. Apart of addressing reviewers' comments, kindly revise the manuscript based on two more points from me.

1. I agree with one of the reviewers, that more details are necessary on how authors calculated the sample size. Kindly provide all assumptions for sample size estimation for future research.

2. The manuscript need careful proofreading. Kindly revise the manuscript carefully for sentence formatting, internal consistency, and punctuation.

Reviewers' comments:

Reviewer's Responses to Questions

**Comments to the Author**

1. Is the manuscript technically sound, and do the data support the conclusions?

Reviewer #1: Yes

Reviewer #2: Partly

2. Has the statistical analysis been performed appropriately and rigorously?

Reviewer #1: Yes

Reviewer #2: Yes

3. Have the authors made all data underlying the findings in their manuscript fully available?

Reviewer #1: Yes

Reviewer #2: Yes

4. Is the manuscript presented in an intelligible fashion and written in standard English?

Reviewer #1: No

Reviewer #2: Yes

Reviewer #1: Thank you for the opportunity to review this manuscript. However, the manuscript would benefit from clearer framing of the research gap, improved methodological clarity, and deeper interpretation of results. My comments below aim to support the authors in enhancing the rigor and clarity of the study.

1. “However, few studies have examined heterogeneity within groups of caregivers of PwD or the characteristics of quality of care, resulting in a lack of empirical basis for early identification and precise support for high-risk caregivers. Notably, with the rapidly growing population of PwD in China, informal caregivers are experiencing an increasingly heavy burden of care.”

While this statement attempts to establish a research gap, it is vague and lacks specificity and supporting evidence. I recommend reframing the gap more precisely, ideally by identifying exactly what has been understudied and how the current study addresses that gap.

2. The statistical analysis section (2.3) demonstrates an appropriate selection of methods, particularly in applying LPA and utilizing relevant model fit criteria. However, the writing suffers from unclear variable definitions, grammatical issues, and imprecise explanations of key statistical concepts. I suggest restructuring the paragraph for clarity, clearly defining all abbreviations, and correcting misleading interpretations (e.g., regarding entropy). These revisions would enhance the section’s clarity and improve its support for the study analytical rigor and reproducibility.

3. The discussion section largely restates that there are three QoC groups, but it does not provide a deep interpretation of what distinguishes these profiles, how meaningful the differences are, or how the findings relate to the theoretical framework.

4. The discussion does not explain why specific predictors (e.g., ADL, pre-illness relationship) influence care quality profiles, nor does it contextualize the findings by comparing them to prior research. A more critical engagement with existing literature would strengthen the discussion and improve the paper contribution to the field.

Reviewer #2: 1. It is unclear which approach ultimately led to the final target of >210 participants, even though the sample size calculation section specifies requirements for both LPA (≥150 participants) and multinomial logistic regression (>200 participants). Could you please clarify if the final target was selected based on the higher of the two estimates or if there was another way that you arrived at the number 213?

2. The sample size calculation assumes 3 latent profiles, but this assumption is not supported by previous research or pilot results. Giving a reference will assure readers of its validity, since the assumed number of profiles directly affects the sample size needed for LPA.

3. It's unclear what is meant by the statement "the mean scores of 11 items of the ECS were used as observed variables." Although it sounds like you averaged the items before performing the analysis, the individual item scores are typically used as the input in latent profile analysis. It appears from Figure 1 that you used the 11-item scores for each individual in the analysis, and then you computed the average scores for each profile to describe them. If this was not what you thought, kindly reword it.

4. The ECS cut-off of 15 is determined by the ROC analysis, and this shall be validated in a different sample, which would improve its clinical applicability across different populations, even though the reported sensitivity and specificity are high.

.

Reviewer #1: No

Reviewer #2: No

---

## [Author Response · Author response to Decision Letter 1]

29 Sep 2025

Response to Editor:

1.I agree with one of the reviewers, that more details are necessary on how authors calculated the sample size. Kindly provide all assumptions for sample size estimation for future research.

Revised

Thanks very much for your review of our manuscript. In response, we have supplemented and refined the description of the sample size estimation accordingly.

For multinomial logistic regression, the sample size should be at least 5-10 times the number of independent variables [24]. With 20 variables included in this study and accounting for a potential 20% rate of invalid questionnaires, a minimum of 120 participants was initially required. To enhance the accuracy and validation efficacy of the potential category model, a sample size of at least 200 was deemed necessary [25]. Initially, 248 participants were recruited. However, 35 were excluded due to incomplete responses or other reasons, resulting in 213 valid participants included in the final analysis (85.89% response rate). (lines 165-173)

[24] Xu, Y., Chen, Z., Tang, X. et al. Latent profile analysis of nutrition knowledge, attitudes, and practices and their influencing factors in maintenance hemodialysis patients. Sci Rep 15, 17246 (2025). https://doi.org/10.1038/s41598-025-02142-4.

[25] Chen X, Wang Z, Wang S, et al. Latent profiles of ambivalence over emotional expression in young breast cancer patients: A cross-sectional study. Int J Nurs Sci. 2025;12(4):379-385. Published 2025 Jun 16. doi:10.1016/j.ijnss.2025.06.005.

2. The manuscript need careful proofreading. Kindly revise the manuscript carefully for sentence formatting, internal consistency, and punctuation.

Revised. We have carefully checked and revised the manuscript to ensure that the sentence format, internal consistency and punctuation are correct.

Response to Reviewer 1:

1. “However, few studies have examined heterogeneity within groups of caregivers of PwD or the characteristics of quality of care, resulting in a lack of empirical basis for early identification and precise support for high-risk caregivers. Notably, with the rapidly growing population of PwD in China, informal caregivers are experiencing an increasingly heavy burden of care.”

While this statement attempts to establish a research gap, it is vague and lacks specificity and supporting evidence. I recommend reframing the gap more precisely, ideally by identifying exactly what has been understudied and how the current study addresses that gap.

Indeedly, while this statement attempts to establish a research gap, it is vague and lacks specificity and supporting evidence. Thank you for your valuable comments. We have revised clearly specify the under-investigated areas within current research and to explicitly elaborate on how our study is designed to address these specific gaps.

Many researches mainly focus on the total score of ECS scale to assess overall QoC and its influencing factors [16-20]. However, these studies adopt a variable-oriented approach, which fails to capture individual differences and obscures heterogeneity among different subgroups. It is essential to identify distinct patterns of the QoC within the group and to implement targeted interventions based on their different characteristics. Although existing research has identified distinct caregiver profiles associated with different types of low QoC in the elderly [21], few studies have examined characteristic within groups of caregivers of PwD, resulting in a lack of empirical basis for precise support for high-risk caregivers of PwD supplied low QoC. (lines 95-104)

[21] Yan E, Lai DWL, Sun R, et al. Typology of family caregivers of older persons: A latent profile analysis using elder mistreatment risk and protective factors. J Elder Abuse Negl. 2023;35(1):34-64. doi:10.1080/08946566.2023.2197269

2. The statistical analysis section (2.3) demonstrates an appropriate selection of methods, particularly in applying LPA and utilizing relevant model fit criteria. However, the writing suffers from unclear variable definitions, grammatical issues, and imprecise explanations of key statistical concepts. I suggest restructuring the paragraph for clarity, clearly defining all abbreviations, and correcting misleading interpretations (e.g., regarding entropy). These revisions would enhance the section’s clarity and improve its support for the study analytical rigor and reproducibility.

Revised.

Thank you for your careful checks. We have made a comprehensive revision to the presentation of the statistical analysis section to ensure clarity and accuracy.

Each item score on the ECS serves as explicit indicators to establish a LPA model using Mplus 8.3. The most suitable model was selected based on fit indices [35], including the Akaike Information Criterion (AIC), Bayesian Information Criterion (BIC), and adjusted Bayesian Information Criterion (aBIC), Lo-Mendell-Rubin likelihood ratio test (LMR), Bootstrapped likelihood ratio test (BLRT), and Entropy. Lower values of AIC, BIC, and aBIC indicated a superior model fit. Differences between latent profile models were compared using the LMR and BLRT. A significant P-value suggests that the model with k classes fits the data better than the model with k-1 classes. Entropy was used to assess classification accuracy, with values closer to 1 indicating more precise classification; an entropy value ≥0.8 indicates a classification accuracy exceeding 90%. Cohen's d was calculated to further validate the accuracy of the classification (0.2-0.5: small; 0.5-0.8: medium; >0.8: large) [36,37]. (lines 230-242)

3. The discussion section largely restates that there are three QoC groups, but it does not provide a deep interpretation of what distinguishes these profiles, how meaningful the differences are, or how the findings relate to the theoretical framework.

Revised.

Thank you very much for this insightful and critical comment. We have thoroughly revised the Discussion section to address these points as follows, analyzing the differences and significance of the three groups of characteristics, as well as linking the results of this study to the theoretical framework.

We used LPA to categorize the caregivers of PwD into three distinct profiles, i.e., high QoC (24.41%), moderate QoC (44.60%), and low QoC (30.99%). Statistically significant differences were found between the high and moderate QoC profile in terms of pre-illness relationship quality and PwD’ ADL. Logistic regression analysis revealed that the high QoC profile was linked to great quality of pre-illness relationship and superior PwD’ ADL. Similarly, significant differences in economic status, perceived overload, and social support were observed between the low and moderate QoC profile. The low QoC profile was associated with low economic affordability, high perceived overload, and inadequate social support. These results highlight that tailored strategies based on the distinct characteristics of each caregiver subgroup. Priority should be given to caregivers facing financial strain, high perceived overload, and insufficient social support. (lines 328-339)

The Extended Stress Process Model hints that background variables, primary stressors, quality of life, and caregiver resources collectively influence the quality of informal care. Our study verified that background variables, primary stressors, and caregiver resources impacted the quality of informal care, except caregivers’ quality of life. Specifically, affordability of living expenses, pre-illness relationship quality, the ADL of PwD, perceived overload, and social support decided three QoC profiles. These findings provide an innovative perspective on the importance of economic status, relationship quality, caregiving stress, social support in mitigating the deterioration of QoC among caregivers of PwD. (lines 340-348)

4. The discussion does not explain why specific predictors (e.g., ADL, pre-illness relationship) influence care quality profiles, nor does it contextualize the findings by comparing them to prior research. A more critical engagement with existing literature would strengthen the discussion and improve the paper contribution to the field.

Revised.

Thanks for your valuable feedback. We have revised to provide a more thorough explanation of how specific predictors (such as Activities of Daily Living and pre-illness relationship quality) influence care quality profiles, and have contextualized our findings through systematic comparisons with previous studies.

Align with previous studies [15,42], a positive pre-illness relationship serves as a protective factor for QoC. As Steadman et al. demonstrated [43], caregivers reporting higher premorbid relationship satisfaction showed significantly less stress and less reactivity to memory and behavioral problems, and better problem solving skills and more effective communication. Conversely, caregivers with poor pre-illness relationships may perceive caregiving responsibilities as an additional burden, leading to negative attitudes and behaviors [41,43], ultimately compromising the quality of informal care. The finding emphasizes the critical role of caregiving context during care, suggesting that interventions aimed at improving QoC should consider pre-illness relationship quality. (lines 352-362)

The decline in ADL among PwD significantly increases caregiver overload [48]. Research has shown that cognitive impairment or psychiatric symptoms are prevalent in most patients at the time of diagnosis [3], substantially compromising their daily functioning and necessitating caregiver assistance. However, many caregivers of PwD struggle to provide consistent and attentive care due to insufficient care capacity and a lack of care resources [49], ultimately resulting in diminished QoC. Furthermore, disability is often accompanied by aphasia and comprehension deficits, making it difficult for patients to express their needs accurately. These challenges may undermine caregivers’ patience and lead to inadequate care. (lines 369-378)

[43] Steadman PL, Tremont G, Davis JD. Premorbid relationship satisfaction and caregiver burden in dementia caregivers. J Geriatr Psychiatry Neurol. 2007;20(2):115-119. doi:10.1177/0891988706298624

[48] Kim, B., Noh, G.O. & Kim, K. Behavioural and psychological symptoms of dementia in patients with Alzheimer’s disease and family caregiver burden: a path analysis. BMC Geriatr 21, 160 (2021). https://doi.org/10.1186/s12877-021-02109-w

[49] Report on the family survival status of Alzheimer's disease patients in China. Alzheimer’s Disease Chinese, 2020-1. 2019. URL: http://weixin.moreedge.cn/bps_2019/index.php

Response to Reviewer 2:

1.It is unclear which approach ultimately led to the final target of >210 participants, even though the sample size calculation section specifies requirements for both LPA (≥150 participants) and multinomial logistic regression (>200 participants). Could you please clarify if the final target was selected based on the higher of the two estimates or if there was another way that you arrived at the number 213?

In accordance with the Editor's suggestion, the revisions have been incorporated. Please see Response to Editor Answer No.1 (lines 165-173 in the revised manuscript).

2.The sample size calculation assumes 3 latent profiles, but this assumption is not supported by previous research or pilot results. Giving a reference will assure readers of its validity, since the assumed number of profiles directly affects the sample size needed for LPA.

Revised.

Thanks the reviewer for this important comment. The assumption regarding the number of potential profiles in the initial submission was not sufficiently grounded. In response, we have removed the unsupported statement and revised.

To enhance the accuracy and validation efficacy of the potential category model, a sample size of at least 200 was deemed necessary [25]. (lines 169-170)

[25] Chen X, Wang Z, Wang S, et al. Latent profiles of ambivalence over emotional expression in young breast cancer patients: A cross-sectional study. Int J Nurs Sci. 2025;12(4):379-385. Published 2025 Jun 16. doi:10.1016/j.ijnss.2025.06.005.

3.It's unclear what is meant by the statement "the mean scores of 11 items of the ECS were used as observed variables." Although it sounds like you averaged the items before performing the analysis, the individual item scores are typically used as the input in latent profile analysis. It appears from Figure 1 that you used the 11-item scores for each individual in the analysis, and then you computed the average scores for each profile to describe them. If this was not what you thought, kindly reword it.

Revised.

We have corrected the incorrect statement in the manuscript.

Each item score on the ECS serves as explicit indicators to establish a LPA model using Mplus 8.3. (lines 230-231)

4. The ECS cut-off of 15 is determined by the ROC analysis, and this shall be validated in a different sample, which would improve its clinical applicability across different populations, even though the reported sensitivity and specificity are high.

Revised.

The original expression of the third limitation has been revised to improve its clarity and precision.

Third, although this study reported high sensitivity and specificity, the ECS cut-off value of 15 was determined through ROC analysis. Future validation in different samples are needed to enhance the clinical applicability of this threshold. (lines 419-421)

---

## [Decision Letter · Decision Letter 1]

15 Dec 2025

Dear Dr. Shi,

Thank you for submitting your manuscript to PLOS ONE. After careful consideration, we feel that it has merit but does not fully meet PLOS ONE’s publication criteria as it currently stands. Therefore, we invite you to submit a revised version of the manuscript that addresses the points raised during the review process.

We look forward to receiving your revised manuscript.

Kind regards,

Arupendra Mozumdar, Ph.D., M.Sc.

Academic Editor

PLOS One

Journal Requirements:

**Additional Editor Comments:**

I went through the rebuttals from the authors and the reviewers' comments on the second version of the manuscript. The manuscript is composed of scores of grammatical errors, awkward sentence structure, and confusing statements. Tow of the reviewers did painstaking yet remarkable jobs to identify the shortcomings and provided valuable comments to improve the manuscript. Keeping the importance of the topic and the level of efforts from the authors in mind I would like to give another chance to the manuscript. Me too have some suggestions to improve the manuscript. I am requesting authors to revise and resubmit the manuscript for another round by addressing all the comments form the reviewers.

The manuscript is probably drafted and written by someone whose first language is not English. So I will eagerly request authors copy-editing the manuscript by someone who is proficient in academic English before submission. My specific comments are given below.

Please provide marginal effect plots or at least a table of mean values along with 95%-CI for independent variables for each of the three groups in the final model. This will provide information to interpret the magnitude and nature of the relationship between predictors and latent profile membership. Instead of just seeing if a variable is a significant predictor, these plots will show how much the probability of profile membership shifts as the independent variable changes.

Reviewer from the last round of review asked to validate ECS cut-off of 15 in a different sample to improve its clinical applicability. I suggest dividing the sample into two sub-samples, say 50-50% or 60-40%, using a random number generator. Then use one sub-sample to ROC analysis get a cut-off, apply the cut-off to other sub-sample, and calculate the sensitivity and specificity.

Line 55-56: Please mention the geographic reference for the expected number of 152 million PwD in 2050. Will the number of 152 million reach worldwide, in Asia, or only in Chaina?

Line 69: Please revise as “…that may negatively impact care recipients’ (CRs) well-being, …”

Line 70: What does it mean by adequacy of caregiving behaviors? The meaning is unclear to me. Kindly reconstruct the sentence.

Line 72-73: Rephrase from “…CRs’ daily basic and instrumental daily needs…” to “…CRs’ basic and instrumental daily needs…”

Line 104: Change the word ‘supplied’. ‘Provided’ would be a better choice of words here.

Line 126: Please clarify whether traditional total score analysis assumes “the same score can represent different qualities” or LPA assumes "the same…”. If the answer is the latter, please rephrase and bring the last clause in the middle of the sentence.

Line 138: Statement #3 is not a hypothesis. To me it is the application of the learnings from hypothesis testing from the first two hypotheses.

Line 98-102: Do authors mean ‘subgroups’ when they use the word group here? I guess authors wanted to make the justification for LPA here, but the communication is not clear and suddenly the introduction of the concept of group confuses the readers.

Line 160-162: The sentence could be rephrased as "Participants with reading difficulties completed the questionnaires with the help of trained research staff.”

Line 167-170: Thank you for addressing this point raised in the previous round of review. The justifications for the sample size are given for analytical sample. However, readers would like to know the representativeness of the sample along with statistical assumptions like expected value of response distribution, level of confidence, margin of error, power, etc.

Figure 1: Please add legends for all three lines. Technically this should not be a line diagram because the variables in the x-axis are completely independent of each other not showing any trend. It would be better if authors leave the dots and remove the lines connecting the data points. Also, I would recommend adding ‘whiskers’, denoting 95%-CIs of item scores, to each dot. This way authors can show how or if the distributions of scores are overlapped or not. But adding whiskers is not mandatory, if that makes the figure more cluttery.

Table 5: Remove the row titled ROC if anything else is not provided.

Line 321. Revise “…life, and caregiver resources predicted…” into “…life, and caregiver resources; and predicted…”

Line 337-338: Authors would need to add a consequence or a finding to this sentence, for example, "...highlight that tailored strategies need to be developed based on the distinct characteristics of each caregiver subgroup.”

Line 343: Kindly reconstruct the sentence. I guess, authors want to say here that the caregivers' quality of life was the sole factor among all independent variable examined that did not show a statistically significant effect on the quality of care provided. One version could be “The study confirmed that background variables, primary stressors, and caregiver resources influenced the quality of informal care, but did not find a similar impact from caregivers’ quality of life.”

Line 348. These findings provide an innovative perspective on the importance of economic status, relationship quality, caregiving stress, and social support in mitigating the deterioration of QoC among caregivers of PwD. Meaning of the sentence is unclear. Kindly rephrase. Do authors mean low QoC by using the word ‘deterioration’? What is ‘innovative’ perspective on importance of such variables, and how this innovation could ‘mitigate’ the deterioration of QoC (or low QoC)?

Table-5 Kindly properly format the table of multinomial regression. The names of the variables are presented as it is entered into the software. Please rewrite the names of the variables in a meaningful way.

Reviewers' comments:

Reviewer's Responses to Questions

**Comments to the Author**

Reviewer #3: (No Response)

Reviewer #4: All comments have been addressed

Reviewer #5: (No Response)

Reviewer #6: All comments have been addressed

2. Is the manuscript technically sound, and do the data support the conclusions?

Reviewer #3: Yes

Reviewer #4: Yes

Reviewer #5: Yes

Reviewer #6: Yes

3. Has the statistical analysis been performed appropriately and rigorously?

Reviewer #3: I Don't Know

Reviewer #4: I Don't Know

Reviewer #5: Yes

Reviewer #6: Yes

4. Have the authors made all data underlying the findings in their manuscript fully available?

Reviewer #3: No

Reviewer #4: Yes

Reviewer #5: (No Response)

Reviewer #6: Yes

5. Is the manuscript presented in an intelligible fashion and written in standard English?

Reviewer #3: No

Reviewer #4: Yes

Reviewer #5: Yes

Reviewer #6: Yes

Reviewer #3: Quality of informal care among informal caregivers of patients with

dementia: a latent profile and ROC analysis

Authors are commended for their superb study focusing on quality of informal care of a vulnerable patient population. Findings not only highlight the profile of the informal caregiver but also, perhaps unintentionally, provides a sense of the predicament of the care recipient.

General comments

The reviewer suggested revision of sentence structures. However, authors are encouraged to thoroughly engage with the manuscript to ensure that all grammatical issues are addressed. Alternatively, authors might consider consulting with a language and/or copy editor.

Abstract

Line 26- 27: the sentence “Additionally, the absence of clearly thresholds for identifying PwD with low quality of informal care poses a challenge for research and clinical practice” should either read “Additionally, the absence of clear thresholds for identifying PwD with low quality of informal care poses a challenge for research and clinical practice” or alternative, “Additionally, the absence of clearly demarcated [or specified] thresholds for identifying PwD with low quality of informal care poses a challenge for research and clinical practice”

Line 27: it is assumed that authors are referring to PwD needs in “… thresholds for identifying PwD with low quality of informal care [needs]?” Or are they referring to identifying PwD receiving “low quality of informal care” or being delivered to?

Line 28: the word “of” seems misplaced in the sentence structure “… aimed to identify potential profiles of among informal caregivers of PwD…”. Furthermore, it is not clear what “potential profiles” authors are referring to? Similarly, it is not clear what authors are referring to with regards to “profile” / “profiles” in the Methods section [i.e line 36]. Furthermore, in the Results section authors refer to quality of care profiles [line 38-39]? It is only after reading the rest of the manuscript that it became clear what “profiles” refer to. This should be remedied.

Line 29: do authors have one profile in mind stating “… explore influencing factors of different profile, …” or should “profile” be revised to read “profiles”?

Line 39-41: it is not clear whether authors are referring to informal carer’s or PwD’s “worse affordability of living expenses, insufficient social support and higher perceived overloaded”?

Line 45: it is not “obvious” which “classification characteristics” authors are referring to.

In general, the abstract does not seem to have been written with the same robustness as the body of the manuscript.

Introduction

Line 99: it might be helpful to clarify which group authors are referring to stating “within the group” for the reader. It is assumed authors are referring to QoC among informal caregivers?

Line 104-115: it is not clear why authors consider reference to LPA, “an individual-centered statistical analysis”, as background versus information related to Materials and Methods? In the event that authors deem reference to LPA necessary in the Introduction section they are encouraged to rephrase LPA-related sentences to meet typical background-related standards. Ultimately, authors need to separate research methodology from practical application / implementation of their findings and move applicable methodology sentences / detail to these sections of the manuscript.

Line 124-130: ultimately reference to statistical analyses applied [(1) … “using LPA …” (2) “… through logistic regression analysis, …” and (3) “… using ROC analysis …”] reflect authors’ choice of statistical analysis with regard to their study and has no bearing on the objectives of the study. Would authors agree that the same objectives could have been addressed using different statistical approaches?

Line 138-139: the reviewer is not convinced that (3) qualifies as a “hypothesis” [line 135]? The sentence rather seems to address “2.3 Statistical analysis”?

Materials and Methods

Line 145: the sentence structure needs to be addressed. It could possibly read “… caregivers attending memory clinics, rehabilitation clinics, and neurological clinics …” or “… caregivers attending either a memory clinic, rehabilitation clinic, and neurological clinic …”.

Line 149: is there a reason why authors reference the ICD-10 [1993] versus the ICD-11 [2019]?

Line 154: what does “clear consciousness” entail? How was “normal intelligence” defined? What criteria did authors apply [and/or how did they gauge] to determine inclusion of informal caregivers based on “clear consciousness, normal intelligence, good communication skills”?

Line 157-174: authors include among these sentences detail regarding a) data collection b) sample size calculation c) ethics approval. It is suggested that authors provide applicable sub-headings to distinguish between various Materials and Method aspects. Streamlining of sub-headings and sentence structures will enhance their paper.

Line 157: it is not clear whether “paper questionnaires” and “independent questionnaires” refer to the ESC caregivers completed or were different questionnaires completed? It might be worth mentioning how many questionnaires [name / label these] participants were expected to complete [possibly in table format] especially given the extended number of questionnaires caregivers were expected to complete [as referred to in 2.2.2]. Such clarity will also assist to grasp the extensive carers profile authors obtained? Did both PwD and the caregivers complete questionnaires independently. Who provided basic information of PwD [line 187]?

Line 160-162: please address the sentence structure as it is not clear what authors intend to convey.

Line 162: does “The questionnaires were distributed and collected immediately” imply that “Participants were expected to complete questionnaires immediately upon distribution.”

Line 167 and 170: while authors reference studies which applied LPA [and indicate sample size calculation [including references] for their respective studies] the reviewer is not convinced that these studies could / should be referenced to corroborate authors’ statements regarding their sample size calculations. Authors are encouraged to preferably source, and reference, (pure) statistics-related publications addressing calculations of sample size (applicable to authors’ study).

Line 180: does Lau, et al. have a reference?

Line 180: the sentence structure should read “ESC was completed ….”

Line 191: the sentence structure should read "Relationship Rewards Scale (RRS) was used to …”. Furthermore, a) the reviewer was unable to source a “Relationship Rewards Scale” and b) use of reference 15 should perhaps be reconsidered as a reference providing the reader to access to RRS.

Results

Line 259: a) “Caregivers was predominantly were female …” should read “Caregivers were predominantly female …” b) please refrain from starting a sentence with a number / percentage. Line 259 could possibly read “Caregivers were predominantly female (64.8%) and 75.6% of the caregivers lived with PwD.”

Table 1: a) it is suggested that authors consider providing a separate table with results based on the different questionnaires caregivers completed. It is debatable whether affordability of living expenses; physical health, depression, quality of pre-illness relationship, perceived overload and social support could / should be considered as “Characteristics” of caregivers. In 2.2.2. authors refer to these as “independent variables” [as reported in line 296-300] b) consider providing a footnote indicating what “Other” relationships with PwD were.

Line 264: revise “… caregivers of PwD were shown …” to read “… caregivers of PwD are shown …”

Table 4: “LPAa” is not explained as a footnote to Table 4.

Figure 1: the only legend regarding colours used corresponds to blue “Low QoC”.

Table 5: the heading of the table indicates the test performed rather than what is reported in Table 5 as a result of the test performed.

Table 7: what is the intention of “ROC” in the table [given the table heading]?

Discussion

Line 319: “Based on Extended Stress Model, …” should read “Based on the Extended Stress Model, …”

Line 321: “… caregiver resources predicted …” should read “… caregiver resources predict ...”

Line 328-336: seems these lines repeat what authors already stated in the Result section and could be better utilized to elicit discussion.

Line 337-338: this sentence is incomplete.

Line 342: provide reference corroborating the statement.

Line 343: it is not clear what authors imply with “… except caregivers’ quality of life”? It seems authors emphasize that the caregiver’s quality of care is impacted but not the person’s quality of life per se?

Line 345: consider revising the word “decided”. It seems authors are stating that the three QoC profiles identified during the study pivoted on / revolved around / was determined by the list of aspects mentioned in the sentence?

Line 381: it is assumed that “… consisting other studies …” should read “… consistent with other studies …”? Irrespective, the sentence does not make sense.

Line 409-411: this sentence seem more fitting as a conclusion instead of a discussion point.

Reviewer #4: Thank you for your paper some comments to consider

People with dementia rather than patients should be used

Background-Whilst this is well written it seems too long and would benefit from some editing

Line 71 pleasant activities ?

Capital letter mid sentence line 101

Lines 360-362 some grammatical issues

Generally just check grammar throughout

Reviewer #5: Good morning.

First, I would like to congratulate the authors on their excellent work and on giving me the opportunity to review it. The topic addressed, the quality of care, is relevant, and the authors' approach provides information that may be useful to the scientific community. The effort made by the authors to convey their results is evident.

As a result of my review, I have the following suggestions regarding your manuscript. My suggestions are divided into general suggestions, which affect the manuscript, and specific suggestions, which are described by line number and section of the manuscript to facilitate their location.

I hope to have the opportunity to review your manuscript again with the suggested corrections.

Reviewer #6: The comments are attached to the document. They are clear and direct and the author can be able to action the issues raised.

.

Reviewer #3: No

Reviewer #4: No

Reviewer #5: **Yes:** Vicente Martín MorenoVicente Martín MorenoVicente Martín MorenoVicente Martín Moreno

Reviewer #6: **Yes:** Moses MutuaMoses MutuaMoses MutuaMoses Mutua

---

## [Author Response · Author response to Decision Letter 2]

21 Feb 2026

Response to Reviewers

Dear Editor and Reviewers,

Thank you for your detailed and constructive comments on my manuscript. According to the editor and reviewers’ comments, we have made extensive corrections to our previous manuscript. Two versions of the revised manuscript are submitted: one with no markings and another with the changes highlighted in yellow for clarity. All authors have reviewed and approved the response letter and the revised manuscript. Below is the point-by-point responses to the editor and reviewers, the red font is the revised original text in the manuscript:

Response to Editor:

1.I went through the rebuttals from the authors and the reviewers' comments on the second version of the manuscript. The manuscript is composed of scores of grammatical errors, awkward sentence structure, and confusing statements. Tow of the reviewers did painstaking yet remarkable jobs to identify the shortcomings and provided valuable comments to improve the manuscript. Keeping the importance of the topic and the level of efforts from the authors in mind I would like to give another chance to the manuscript. Me too have some suggestions to improve the manuscript. I am requesting authors to revise and resubmit the manuscript for another round by addressing all the comments form the reviewers. The manuscript is probably drafted and written by someone whose first language is not English. So I will eagerly request authors copy-editing the manuscript by someone who is proficient in academic English before submission.

Thank you for the decision and feedback. We will have the manuscript professionally copy-edited and fully address all reviewer comments in the revision.

2.Please provide marginal effect plots or at least a table of mean values along with 95%-CI for independent variables for each of the three groups in the final model. This will provide information to interpret the magnitude and nature of the relationship between predictors and latent profile membership. Instead of just seeing if a variable is a significant predictor, these plots will show how much the probability of profile membership shifts as the independent variable changes.

Revised.

Thanks very much for your review of our manuscript. Revised as suggested. We have added the OR and 95% CI for independent variables in Table 6. (line 313)

3.Reviewer from the last round of review asked to validate ECS cut-off of 15 in a different sample to improve its clinical applicability. I suggest dividing the sample into two sub-samples, say 50-50% or 60-40%, using a random number generator. Then use one sub-sample to ROC analysis get a cut-off, apply the cut-off to other sub-sample, and calculate the sensitivity and specificity.

Revised.

Thank you for this important methodological suggestion. We have followed the recommended procedure: randomly splitting the sample (50-50%), deriving the cut-off in one subsample, and validating it in the other. The analysis robustly confirmed the ECS cut-off of 15 in both independent subsamples, supporting its reliability. The detailed results have been added to the manuscript.

To validate the clinical applicability of the ECS cut-off, the sample was randomly split into two independent subsamples (approximately 50%/50%). In the derivation subsample, ROC analysis identified an optimal ECS cut-off score of 15, yielding a high sensitivity of 0.993, specificity of 0.955, and an AUC of 0.998 (95% CI [0.993, 0.999], P < 0.001; Fig. 2; Table 7). The mean difference between the two groups was statistically significant with a large effect size (Table 8), further supporting this cut-off point. In the validation subsample, the identical cut-off of 15 was confirmed, achieving a Youden’s index of 0.928, sensitivity of 0.985, specificity of 0.943, and an AUC of 0.990 (95% CI [0.975, 0.999], P < 0.001). The consistent performance across both subsamples supports the robustness and generalizability of the ECS cut-off score of 15 for identifying low QoC. (line 316-326)

4.Line 55-56: Please mention the geographic reference for the expected number of 152 million PwD in 2050. Will the number of 152 million reach worldwide, in Asia, or only in China?

Revised.

Thank you for your valuable comments. We have revised clearly specify the data "152 million in 2050" refers to the global level.

Globally, the number of people with dementia (PwD) is projected to reach 152 million by 2050 [2]. (line 55-56)

5.Line 69: Please revise as "…that may negatively impact care recipients’ (CRs) well-being, …"

Revised. (line 70)

6.Line 70: What does it mean by adequacy of caregiving behaviors? The meaning is unclear to me. Kindly reconstruct the sentence.

Revised.

Thank you for pointing out this issue. In the revised manuscript, we explicitly define "adequacy of caregiving behaviors" as "the adequacy of caregiving behaviors in meeting the CRs’ basic needs". (line 71)

7.Line 72-73: Rephrase from "…CRs’ daily basic and instrumental daily needs…" to "…CRs’ basic and instrumental daily needs…"

Revised. (line 74-75)

8.Line 104: Change the word ‘supplied’. ‘Provided’ would be a better choice of words here.

Revised. Thank you for your suggestion. I have revised ‘supplied’ to ‘provided’ and further refined the sentence for clarity and academic tone. The revised sentence now reads:

This gap limits the empirical basis for providing targeted support to high-risk caregivers who provide low QoC for PwD. (line 104-105)

9.Line 126: Please clarify whether traditional total score analysis assumes "the same score can represent different qualities" or LPA assumes "the same…". If the answer is the latter, please rephrase and bring the last clause in the middle of the sentence.

Thank you for this correct observation. We agree that the phrase was ambiguously phrased and that the limitation belongs to the traditional score approach. As the specific sentence has been revised following thet suggestion of Reviewer 3 (to state objectives without specifying methods). Please see Response to Reviewer 3 Answer No.10 (line 127-132 in the revised manuscript).

10.Line 138: Statement #3 is not a hypothesis. To me it is the application of the learnings from hypothesis testing from the first two hypotheses.

Revised.

Thank you for your precise feedback. We agree that the original statement described an application. Following Reviewer 6’s suggestion, we have revised hypothesis: (3) On this basis, and through ROC analysis, it may be possible to establish an optimal cutoff value with high sensitivity and specificity that allows for the identification of different levels of care quality and, specifically, the identification of low-quality care. (line119-122)

11.Line 98-102: Do authors mean ‘subgroups’ when they use the word group here? I guess authors wanted to make the justification for LPA here, but the communication is not clear and suddenly the introduction of the concept of group confuses the readers.

Thank you for this clarification. We have revised the paragraph to address the ambiguity. The text now clearly distinguishes the limitations of the traditional variable-centered approach (focusing on total scores) from the need for a person-centered approach (to identify distinct profiles).

A shift to a person-centered approach is therefore needed to identify distinct caregiver profiles for personalized support. Although such person-centered methods have identified meaningful profiles in elderly care [21], their application specifically to dementia caregivers remains scarce. (line101-104)

12.Line 160-162: The sentence could be rephrased as "Participants with reading difficulties completed the questionnaires with the help of trained research staff."

Revised. (line 165-166)

13.Line 167-170: Thank you for addressing this point raised in the previous round of review. The justifications for the sample size are given for analytical sample. However, readers would like to know the representativeness of the sample along with statistical assumptions like expected value of response distribution, level of confidence, margin of error, power, etc.

Revised.

Thank you for the valuable comments.We have conducted an a priori power analysis using G*Power and revised the manuscript accordingly, clarifying both the calculation basis and the methodological considerations for determining the sample size.

An a priori sample size calculation was performed using G*Power 3.1.9.7. With a two-tailed, a medium effect size (f2=0.30), a power of 0.95, a statistical level α=0.05 [25], and 20 predictors, the minimum required sample size was 120. To ensure robust model fit, stable parameter estimation, and adequate power for subgroup comparisons in LPA, we targeted a minimum of 200 participants [26], consistent with methodological recommendations for person-centered approaches. (line171-176)

14.Figure 1: Please add legends for all three lines. Technically this should not be a line diagram because the variables in the x-axis are completely independent of each other not showing any trend. It would be better if authors leave the dots and remove the lines connecting the data points. Also, I would recommend adding ‘whiskers’, denoting 95%-CIs of item scores, to each dot. This way authors can show how or if the distributions of scores are overlapped or not. But adding whiskers is not mandatory, if that makes the figure more cluttery.

Revised.

We add legends for all three lines. We leave the dots and remove the lines connecting the data points and adding ‘whiskers’. (line 299)

15.Table 5: Remove the row titled ROC if anything else is not provided.

Thank you for your comment. We have checked Table 5 (now table 6) carefully and found no row titled "ROC" in the current version. The table includes only the variables listed in the submitted manuscript. Could you please clarify which "ROC" you are referring to? We are happy to make the necessary changes once we understand your request.

16.Line 321. Revise "…life, and caregiver resources predicted…" into "…life, and caregiver resources; and predicted…"

Thank you for your suggestion. I agree with the proposed revision. However, in response to Reviewer 6’s comment on content redundancy, the sentence cited at Line 321 has been removed during revision. The relevant section no longer appears in the manuscript.

17.Line 337-338: Authors would need to add a consequence or a finding to this sentence, for example, "...highlight that tailored strategies need to be developed based on the distinct characteristics of each caregiver subgroup."

Revised. Thank you for your suggestion on Lines 337-338. I agree that a clearer implication is needed. In response to Reviewer 6’s comments, I have rewritten this section.

18.Line 343: Kindly reconstruct the sentence. I guess, authors want to say here that the caregivers' quality of life was the sole factor among all independent variable examined that did not show a statistically significant effect on the quality of care provided. One version could be "The study confirmed that background variables, primary stressors, and caregiver resources influenced the quality of informal care, but did not find a similar impact from caregivers’ quality of life."

Revised.

Overall, our findings are largely consistent with Extended Stress Process Model [16], confirming that background variables, primary stressors, and caregiver resources significantly predict the quality of informal care,but did not find a similar impact from caregivers’ quality of life. (line 357-360)

19.Line 348. These findings provide an innovative perspective on the importance of economic status, relationship quality, caregiving stress, and social support in mitigating the deterioration of QoC among caregivers of PwD. Meaning of the sentence is unclear. Kindly rephrase. Do authors mean low QoC by using the word ‘deterioration’? What is ‘innovative’ perspective on importance of such variables, and how this innovation could ‘mitigate’ the deterioration of QoC (or low QoC)?

Thank you for your comment on Line 348. I agree the original sentence was unclear. In response to Reviewer 6’s comment that the Discussion should not present new results, I have rewritten this section.The relevant section no longer appears in the manuscript.

20.Table-5 Kindly properly format the table of multinomial regression. The names of the variables are presented as it is entered into the software. Please rewrite the names of the variables in a meaningful way.

Thank you for your suggestion. I have revised Table 5 (now Table 6) and rewritten all variable names to ensure they are presented clearly and meaningfully. (line 313)

Response to Reviewer 3:

Authors are commended for their superb study focusing on quality of informal care of a vulnerable patient population. Findings not only highlight the profile of the informal caregiver but also, perhaps unintentionally, provides a sense of the predicament of the care recipient.

General comments:

The reviewer suggested revision of sentence structures. However, authors are encouraged to thoroughly engage with the manuscript to ensure that all grammatical issues are addressed. Alternatively, authors might consider consulting with a language and/or copy editor.

Revised. We have carefully checked and revised revision of sentence structures, thoroughly engage with the manuscript to ensure that all grammatical issues are addressed.

一．Abstract

1.Line 26-27: the sentence "Additionally, the absence of clearly thresholds for identifying PwD with low quality of informal care poses a challenge for research and clinical practice" should either read "Additionally, the absence of clear thresholds for identifying PwD with low quality of informal care poses a challenge for research and clinical practice" or alternative, "Additionally, the absence of clearly demarcated [or specified] thresholds for identifying PwD with low quality of informal care poses a challenge for research and clinical practice"

Revised.

Additionally, the absence of clear thresholds to identify PwD receiving low-quality informal care poses a challenge for research and clinical practice. (line 25-27)

2.Line 27: it is assumed that authors are referring to PwD needs in "… thresholds for identifying PwD with low quality of informal care [needs]?" Or are they referring to identifying PwD receiving "low quality of informal care" or being delivered to?

Revised. Thank you for the feedback. We have revised the sentence to eliminate the ambiguity.

Additionally, the absence of clear thresholds to identify PwD receiving low-quality informal care poses a challenge for research and clinical practice. (line 25-27)

3.Line 28: the word "of" seems misplaced in the sentence structure "… aimed to identify potential profiles of among informal caregivers of PwD…". Furthermore, it is not clear what "potential profiles" authors are referring to? Similarly, it is not clear what authors are referring to with regards to "profile" / "profiles" in the Methods section [i.e line 36]. Furthermore, in the Results section authors refer to quality of care profiles [line 38-39]? It is only after reading the rest of the manuscript that it became clear what "profiles" refer to. This should be remedied.

Revised.

Thank you for this essential critique regarding the unclear introduction of a core concept. We have remedied this by providing a clear and full definition at the first occurrence of the term “profiles” in the Introduction (Line 28), specifying that they refer to “profiles of quality of care (QoC)”. Subsequently, in the Methods and Results sections, we consistently use this defined term (“QoC profiles”) to ensure clarity and avoid ambiguity.

Thus, this study aimed to identify the profiles of quality of care (QoC) among informal caregivers of PwD, explore influencing factors of different profile, and determine the optimal cut-off value of the Exemplary Care Scale (ECS). (line 27-29)

4.Line 29: do authors have one profile in mind stating "… explore influencing factors of different profile, …" or should "profile" be revised to read "profiles"?

Revised. The term of "profile" was revised to "profiles".

5.Line 39-41: it is not clear whether authors are referring to informal carer’s or PwD’s "worse affordability of living e

---

## [Editor Report · Decision Letter 2]

3 Mar 2026

Quality of informal care among informal caregivers of people with dementia : a latent profile and ROC analysis

PLOS One

Dear Dr. Shi,

Thank you for submitting your manuscript to PLOS ONE. After careful consideration, we feel that it has merit but does not fully meet PLOS ONE’s publication criteria as it currently stands. Therefore, we invite you to submit a revised version of the manuscript that addresses the points raised during the review process.

We look forward to receiving your revised manuscript.

Kind regards,

Arupendra Mozumdar, Ph.D., M.Sc.

Academic Editor

PLOS One

Journal Requirements:

Additional Editor Comments:

Thank you very much for submitting the revised manuscript titled “Quality of informal care among informal caregivers of people with dementia : a latent profile and ROC analysis” after incorporating the suggestions from the editor and the reviewers. After carefully going through the revised manuscript and the responses to reviewers’ comments, I still have some concerns, addressing those, I believe, would enhance the quality of the manuscript up to the standard of publication for the journal.

I would like to request the authors to strictly follow the instructions for authors while preparing tables and figures, and how the in-text citation to those tables and figures have been made. Many of the tables misses the key elements in titles, footnotes, and legends. We would not be able to accept a manuscript that which has not been prepared strictly following the instructions to authors. You may consider of taking help from the experts for preparing the manuscript.

Some tables in the appendices seem redundant. The table numbers in the manuscript and the appendices are overlapping, titles and notes are incomplete, and without any reference to the text of the manuscript. Some column headings are given in complete lower case. Requesting authors to be careful while preparing the manuscript even if those are supporting materials. Appendices are not meant for dumping of information and analyses, but to provide readers the context without confusing them.

Method: Sample size calculation. Please remove the response rate from the paragraph. I believe all 248 pair of caregivers and PwD responded to the study, but the current analyses included data from 213 out of 248 pairs. This is not the response rate as we experience in survey where not all persons are approached may not respond.

Fig 2. Please mention the name of the score / scale, of which the cut-off of 15 was estimated, in the title of ROC curve.

Appendix-2 title of the ROC curve. Font not in English. No title of the figure. The ROC curve looks completely different what is provided in the manuscript (Fig 2). What is the full form of AUC.

Table 8. Please include the name of the variable in the title of the table (like you did for Table 5) of which the mean and SD have been calculated.

Specifically, contrary to both the model and prior research [7,16,41], caregiver depression, a core dimension of quality of life, did not show a significant association with QoC.

Further, while explaining such non-significant association between depression and QoC author questioned the validity of the scale of depression in Chinese population, and inferred that the depression may be under-reported and that underreporting could be due to fear of social disapproval or strong internalized expectations among Chinese culture.

At this point there are two situations. Either, the depression scale is not valid for the Chinese population, and we should not use the variable at all, or may be the non-significant result is the significance of the association. May be the caregivers are providing good quality of care to the PwD, as it is the cultural practice to care about them, in spite of themselves feeling depressed in providing care due to different socioeconomic constraints.

Adding to my surprise, Table 1 of Appendix 1 showed a highly significant association of depression (p< 0.001) across three latent profile groups. Further, the manuscript also reported that “Notably, lower income was also significantly associated with higher depressive symptoms, suggesting that financial strain undermines psychological well-being and further compounds care quality risks. These findings therefore identify caregivers with low income, low education, rural residence, and elevated depressive symptoms as a multiple disadvantaged subgroup, underscoring the need for integrated economic, mental health, and educational support from social services.”

So, I am getting confused what is the actual finding of the paper related to QoC and depression.

---

## [Author Response · Author response to Decision Letter 3]

18 Mar 2026

Response to Editor

Dear Editor,

Thank you for your detailed and constructive comments on my manuscript. According to the editor and reviewers’ comments, we have made extensive corrections to our previous manuscript. Two versions of the revised manuscript are submitted: one with no markings and another with the changes highlighted in yellow for clarity. Below are our point-by-point responses to the editor, the red font is the revised original text in the manuscript:

Journal Requirements:

Thank you for this important reminder. We have carefully reviewed the entire reference list to ensure its completeness and accuracy. All in-text citations have been verified against the reference list to confirm that no citations are missing and that all listed references are cited in the text. We have consulted PubMed to confirm that none of the cited papers have been retracted. All necessary corrections have been made in the revised manuscript, and we believe the reference list now meets the journal's standards.

Additional Editor Comments:

1. Thank you very much for submitting the revised manuscript titled “Quality of informal care among informal caregivers of people with dementia : a latent profile and ROC analysis” after incorporating the suggestions from the editor and the reviewers. After carefully going through the revised manuscript and the responses to reviewers’ comments, I still have some concerns, addressing those, I believe, would enhance the quality of the manuscript up to the standard of publication for the journal.

Thank you very much for the opportunity to further revise our manuscript. We have carefully addressed all the remaining concerns raised by you and the reviewers. The point-by-point responses are provided below, and all revisions are clearly marked in the manuscript. We believe the revised manuscript is now suitable for publication and look forward to your final decision.

2. I would like to request the authors to strictly follow the instructions for authors while preparing tables and figures, and how the in-text citation to those tables and figures have been made. Many of the tables misses the key elements in titles, footnotes, and legends. We would not be able to accept a manuscript that which has not been prepared strictly following the instructions to authors. You may consider of taking help from the experts for preparing the manuscript.

Thank you very much for your valuable feedback. We sincerely apologize for the formatting issues in our tables and figures. We have now carefully reviewed the journal’s “Instructions for Authors” and revised all tables and figures accordingly. Specifically, we have: Added complete titles to all tables and figures. Included necessary footnotes and legends to explain abbreviations and symbols. Corrected in-text citations to ensure they follow the journal’s style. We also sought assistance from a colleague experienced in manuscript preparation to verify compliance. We believe the revised manuscript now fully meets the journal’s formatting standards.

3. Some tables in the appendices seem redundant. The table numbers in the manuscript and the appendices are overlapping, titles and notes are incomplete, and without any reference to the text of the manuscript. Some column headings are given in complete lower case. Requesting authors to be careful while preparing the manuscript even if those are supporting materials. Appendices are not meant for dumping of information and analyses, but to provide readers the context without confusing them.

Thank you for your detailed review of the appendices. We have carefully revised all supplementary materials according to your suggestions:

First, to avoid numbering overlap with the main text, the remaining appendices have been renumbered as S1 Table, S2 Table, and S1 Fig, with corresponding citations now added in the main manuscript.

Second, complete titles and detailed notes have been added to each table and figure, and all column headings have been corrected to proper case.

Third, regarding Appendix 2, we acknowledge that this appendix, which was created to address specific comments from Reviewer 6, did not provide meaningful information relevant to the main findings. Since these analyses are not cited in the main text, we have decided to remove Appendix 2 entirely to streamline the supplementary materials and avoid redundancy.

4. Method: Sample size calculation. Please remove the response rate from the paragraph. I believe all 248 pair of caregivers and PwD responded to the study, but the current analyses included data from 213 out of 248 pairs. This is not the response rate as we experience in survey where not all persons are approached may not respond.

Thank you for this important clarification. We agree that “response rate” is not the appropriate term in this context. We have revised the sentence to accurately reflect that the 213 dyads represent the final analytic sample. The revised text now reads:

Initially, 248 caregiver-PwD dyads were recruited. After excluding 35 dyads due to incomplete responses or other reasons, 213 valid dyads were included in the final analysis. (line 176-178)

5.Fig 2. Please mention the name of the score / scale, of which the cut-off of 15 was estimated, in the title of ROC curve.

Thank you for your suggestion. We have revised the title of Fig. 2 to clearly indicate the name of the scale used to estimate the cut-off value of 15. The revised title now reads: "Fig. 2. ROC curve of the Exemplary Care Scale for classifying low and high quality of care." (line 324)

6.Appendix-2 title of the ROC curve. Font not in English. No title of the figure. The ROC curve looks completely different what is provided in the manuscript (Fig 2). What is the full form of AUC.

Thank you for reviewing Appendix 2. This ROC curve was created in response to Reviewer 6's suggestion (Comment #48) to explore whether the ECS cutoff differs by sex. We conducted sex-specific ROC analyses using QoC as the classification variable. However, classification performance was poor (The area under the curve, AUC = 0.527), and no valid sex-specific cutoff could be established. Since these analyses did not provide meaningful information relevant to our main findings, and to avoid redundancy as per your guidance, we have decided to remove Appendix 2 entirely from the revised manuscript.

7. Table 8. Please include the name of the variable in the title of the table (like you did for Table 5) of which the mean and SD have been calculated.

Thank you for your guidance. The change has been marked in the revised manuscript. Table 8. Negative and positive groups classified by the optimal cut-off score of the Exemplary Care Scale. (line 325)

8. Specifically, contrary to both the model and prior research [7,16,41], caregiver depression, a core dimension of quality of life, did not show a significant association with QoC. Further, while explaining such non-significant association between depression and QoC author questioned the validity of the scale of depression in Chinese population, and inferred that the depression may be under-reported and that underreporting could be due to fear of social disapproval or strong internalized expectations among Chinese culture. At this point there are two situations. Either, the depression scale is not valid for the Chinese population, and we should not use the variable at all, or may be the non-significant result is the significance of the association. May be the caregivers are providing good quality of care to the PwD, as it is the cultural practice to care about them, in spite of themselves feeling depressed in providing care due to different socioeconomic constraints.

Thank you for this insightful observation. We acknowledge that there was a lack of clarity regarding the role of depression, which created confusion. We agree with the reviewer that simply dismissing the depression scale as invalid would be inconsistent with the data. We have softened this claim. Instead, we now argue that cultural factors may lead to either under-reporting or a decoupling of depressive feelings from caregiving behaviors. We have now substantially revised the Discussion section to address this:

This nonsignificant association can be attributed to Chinese cultural norms, particularly filial piety and family responsibility. This cultural value has been linked to reduced caregiver burden and enhanced QoC [42], as home-based dementia care in China is rooted in familism and filial norms [43]. Within this context, family caregivers may find fulfillment in their caregiving role [44], reinforced by community recognition, which enables them to sustain caregiving standards despite experiencing depressive symptoms. This mechanism explains our core finding: although depressive symptoms differed significantly across the three latent profile groups (S1 Table), they did not predict QoC. In other words, depression effectively captured caregiver heterogeneity, yet its impact on caregiving behavior was buffered by cultural responsibility. Thus, the observed nonsignificant association reflects the unique cultural context of family caregiving in China, rather than any inadequacy in scale validity. Future research should develop culturally sensitive models for assessing QoC. (line 355-368)

---

## [Editor Report · Decision Letter 3]

22 Mar 2026

Quality of informal care among informal caregivers of people with dementia : a latent profile and ROC analysis

PONE-D-25-38583R3

Dear Dr. Shi,

We’re pleased to inform you that your manuscript has been judged scientifically suitable for publication and will be formally accepted for publication once it meets all outstanding technical requirements.

Kind regards,

Arupendra Mozumdar, Ph.D., M.Sc.

Academic Editor

PLOS One

Additional Editor Comments (optional):

I congratulate the authors for your handwork and resilient efforts they put behind the manuscript. Authors made satisfactory revisions to their manuscript by incorporating the suggestions put forward by the reviewers and editor. Thereby, I strongly recommend the paper to be accepted for the publication in the journal.
---

## [Editor Report · Acceptance letter]

PONE-D-25-38583R3

PLOS One

Dear Dr. Shi,

I'm pleased to inform you that your manuscript has been deemed suitable for publication in PLOS One. Congratulations! Your manuscript is now being handed over to our production team.

Kind regards,

on behalf of

Dr. Arupendra Mozumdar

Academic Editor

PLOS One